# Transgenic Microalgae Expressing Double-Stranded RNA as Potential Feed Supplements for Controlling White Spot Syndrome in Shrimp Aquaculture

**DOI:** 10.3390/microorganisms11081893

**Published:** 2023-07-26

**Authors:** Patai Charoonnart, Henry Nicholas Taunt, Luyao Yang, Conner Webb, Colin Robinson, Vanvimon Saksmerprome, Saul Purton

**Affiliations:** 1Center of Excellence for Shrimp Molecular Biology and Biotechnology (Centex Shrimp), Faculty of Science, Mahidol University, Bangkok 10400, Thailand; patai.cha@biotec.or.th (P.C.); vanvimon.sak@biotec.or.th (V.S.); 2National Center for Genetic Engineering and Biotechnology (BIOTEC), National Science and Technology Development Agency, Pathum Thani 12120, Thailand; 3Department of Structural and Molecular Biology, University College London, London WC1E 6BT, UK; 4Centre for Molecular Processing, School of Biosciences, University of Kent, Canterbury CT2 7NJ, UK

**Keywords:** *Chlamydomonas reinhardtii*, chloroplast engineering, double-stranded RNA, RNA interference, shrimp aquaculture, white spot syndrome virus

## Abstract

Viral infection of farmed fish and shellfish represents a major issue within the aquaculture industry. One potential control strategy involves RNA interference of viral gene expression through the oral delivery of specific double-stranded RNA (dsRNA). In previous work, we have shown that recombinant dsRNA can be produced in the chloroplast of the edible microalga *Chlamydomonas reinhardtii* and used to control disease in shrimp. Here, we report a significant improvement in antiviral dsRNA production and its use to protect shrimp against white spot syndrome virus (WSSV). A new strategy for dsRNA synthesis was developed that uses two convergent copies of the endogenous *rrnS* promoter to drive high-level transcription of both strands of the WSSV gene element in the chloroplast. Quantitative RT-PCR indicated that ~119 ng dsRNA was produced per liter of culture of the transgenic microalga. This represents an ~10-fold increase in dsRNA relative to our previous report. The engineered alga was assessed for its ability to prevent WSSV infection when fed to shrimp larvae prior to a challenge with the virus. The survival of shrimp given feed supplemented with dried alga containing the dsRNA was significantly enhanced (~69% survival) relative to a negative control (<10% survival). The findings suggest that this new dsRNA production platform could be employed as a low-cost, low-tech control method for aquaculture.

## 1. Introduction

White Spot Syndrome Virus (WSSV) is the causative agent of White Spot Syndrome Disease (WSSD) in penaeid shrimp, including commercially relevant species such as black tiger prawns, king prawns, and Atlantic white shrimp. The virus was first identified during a 1992 outbreak in mainland China and Taiwan, where it is still endemic in shrimp cultivation, and WSSD is widely regarded as the most detrimental disease seen in commercial aquaculture of these organisms, causing up to 100% mortality within 3–5 days [1].

Chemical agents that can be deployed to combat such viral infections are significantly limited compared to the wide range of antibiotics that can be used to manage bacterial pathogens. This has led to considerable research efforts aimed at developing mitigation strategies for WSSD, with several biotechnological measures reported that decrease or delay losses in shrimp populations under lab-scale conditions [2]. One potential strategy involves RNA interference (RNAi), which has been reported to provide a high degree of protection against WSSV infection [3]. Intramuscular or oral delivery of double-stranded RNA (dsRNA) into the shrimp tissue triggers a systemic RNAi mechanism. If the dsRNA sequence corresponds to a region of a viral gene, then RNAi results in the specific degradation of the mRNA for that gene, thus halting infection. In terms of gene targets, a key mediator of WSSV entry into the host cell is the major envelope protein VP28 [4]. The *VP28* gene is therefore an attractive target for RNAi, and specific silencing of *VP28* by the injection of short interfering RNAs has been shown to reduce viral copy number in infected shrimp, resulting in a reduction in shrimp mortality [5]. In a later study, the feeding of shrimp with *Escherichia coli* expressing *VP28* dsRNA yielded survival rates up to 80% relative to untreated shrimp [6]. Although such oral delivery of dsRNA encapsulated in *E. coli* has shown promise, the strategy does not lend itself to commercial application, not least because of concerns over the dietary effects of *E. coli* on the shrimp, and the potential for horizon gene transfer to pathogenic bacteria of the antibiotic resistance genes contained within the engineered *E. coli*. What is required is a dsRNA production host that is benign and a natural part of the diet of juvenile shrimp, and can be genetically engineered without the requirement for selection using antibiotic-resistance genes.

Microalgae represent one such host [7] and there has been a significant rise in interest in using edible microalgal species as food and feed ingredients over the past two decades, largely due to their sustainable nature and rich nutritional profile [8]. Examples of commercially available microalgae for human consumption include *Chlorella vulgaris*, *Dunaliella salina*, *Haematococcus pluvialis*, and *Euglena gracilis*, with many other species forming an important component of animal feeds, including for shrimp and fish larvae [9]. DNA transformation of the nuclear genome is now possible for numerous microalgal species, with a smaller subset also amenable to chloroplast transformation [10,11]. One application of these technologies is therefore the recombinant production of functional biomolecules for oral delivery to aquatic animals [12].

*Chlamydomonas reinhardtii*, a freshwater microalga, is arguably the most studied microalgal species, and as such has an advanced set of molecular tools for both nuclear [13] and chloroplast genetic engineering [14]. *C. reinhardtii* is also capable of heterotrophic growth if provided with a fixed carbon source such as acetate, allowing the isolation of viable mutants defective in photosynthesis [15]. This has not only made the microalga a model organism for studying photosynthesis, but also allows endogenous photosynthetic genes to be used as selectable markers for the restoration of phototrophic growth, thereby side-stepping traditional markers based on antibiotic resistance genes. Here a wild-type copy of the photosynthetic gene is introduced into the corresponding *C. reinhardtii* mutant and rescued transformants are selected by plating on minimal medium [13,16]. Furthermore, the GRAS (General Recognized As Safe) status of *C. reinhardtii* and ease of chloroplast transformation make it an attractive chassis for production of biomolecules that accumulate in the chloroplast, are bioencapsulated when the cells are dried, and can be delivered orally to the shrimp [17]. We previously reported the first use of this system for the production of dsRNA targeting yellow head virus, and demonstrated partial protection against the virus when administered to *Penaeus vannamei* shrimp [18]. The lack of full protection was attributed to the relatively low level of antiviral dsRNA present in the transgenic *C. reinhardtii* culture (i.e., 16 ng/L; [18]). The present study set out to remedy this shortfall by increasing the level of dsRNA produced in the algal chloroplast using a novel expression cassette. We demonstrate the efficacy of the improved platform both by direct quantification of a dsRNA product targeting the *VP28* gene of WSSV, and by the protection afforded to treated shrimp when challenged with WSSV.

## 2. Materials and Methods

### 2.1. Algal Strain and Maintenance

The *C. reinhardtii* strain CC-5168 (*cw15*, Δ*psbH*), also known as TN72, was used as a recipient line [19]. The strain was maintained on Tris Acetate Phosphate (TAP) medium [20] solidified with 2% agar, and grown under continuous low-intensity light (<10 µmol photon m^−2^ s^−1^) at 25 °C. Sub-culturing was performed monthly. Two full loops of healthy *C. reinhardtii* from solid medium were used to inoculate 400 mL TAP for liquid cultivation.

### 2.2. Plasmid Design and Construction

The p2xTRBL dsRNA expressing cassette was constructed using a Golden Gate approach modeled on the Start–Stop assembly method [21]. Three Level 0 plasmids were built containing, respectively: the 5′ *atpB*_Terminator/*rrnS*_promoter element, the cassette encoding the monomeric Red Fluorescent Protein (mRFP), and the 3′ r*rnS*_promoter/*rbcL*_terminator element. The *C. reinhardtii* expression elements were amplified from genomic chloroplast DNA, with the addition of adaptor sequences to allow scar-free fusion of the promoter/terminator pair, and insertion into the Level 0_Empty recipient vector. Also included were additional adapter sequences to allow the three Level 0 constructs to be assembled into the final Level 1 vector, p2xTRBL. The mRFP cassette was synthesized *de novo*, incorporating both adapter sequences for assembly, and additional sequences to allow dsRNA target sequences to be cloned into the final vector. Construction of this vector is summarized in Appendix A with the plasmid map shown in Appendix A.

A 307 bp fragment of the WSSV VP28 gene (GenBank: DQ979320) was amplified by PCR from a crude virus preparation with primers designed to include inward-facing *Esp*3I sites and specific fusion sites for cloning into p2xTRBL (see Appendix A). The Golden Gate reaction was conducted with 1× CutSmart buffer, 1 mM ATP, 400 units T4 ligase, and 10 units *Esp*3I at a plasmid-to-fragment molar ratio of 1:8. The reactions were incubated at 37 °C for 15 min, followed by 30 cycles of 16/37 °C (2 min each), with a final digestion step at 37 °C for 15 min, and heat inactivation at 65 °C for 20 min. The resultant products were transformed into DH5α (Thermo Fisher Scientific, Waltham, MA, USA) and plated onto LB medium containing 10 µg/mL tetracycline. White colonies were picked and the correct insertion of the WSSV fragment was confirmed by PCR. The resulting plasmid was named p2xTRBL-VP28. The dsRNA expression cassette from this plasmid was then sub-cloned into the *C. reinhardtii* chloroplast transformation vector pSS116 (Appendix A) following a similar protocol but using the dual outward-facing *Bsa*I sites, and with transformants plated on LB medium containing 100 µg/mL ampicillin (LB + Amp^100^). White colonies were again confirmed by PCR, with the final construct named pSS116-VP28. Additional confirmation of correct cloning was achieved by diagnostic restriction endonuclease digestion using *Xho*I and *Bst*XI. The sequence of the *VP28* transcriptional cassette in pSS116-VP28 is shown in Appendix A. This plasmid was amplified in *E. coli* DH5α for *C. reinhardtii* transformation.

### 2.3. Confirmation of dsRNA Expression in E. coli

Plasmid pSS116-VP28 was transformed into the RNase III deficient *E. coli* strain HT115(DE3) (Gentaur Europe, Kampenhout, Belgium; [22]) and colonies were selected on LB + Amp^100^. A single colony was inoculated into liquid LB + Amp^100^ broth and continuously shaken at 37 °C for 18 h. Re-inoculation was performed at 1:100 into fresh LB + Amp^100^ broth and continuously shaken for 8 h followed by cell collection by centrifugation. Total RNA was extracted using the phenol-chloroform (1:1) extraction method, and dsRNA was purified according to [18]. Purified dsRNA was visualized by agarose gel electrophoresis and kept at –20 °C for further use as a standard for the detection of dsRNA from microalgal transformants.

### 2.4. Chloroplast Transformation and Transformant Selection

Chloroplast transformation of the cell wall-deficient strain TN72 was performed using the glass bead transformation method [23]. Briefly, 10 mL of starter culture was inoculated into 400 mL TAP medium and cultured with continuous shaking under low-intensity light until late log-phase (OD_750_ at 1.0–1.2). Cells were then concentrated 100-fold to obtain a final cell density at approximately 2 × 10^8^ cells/mL. Then, 300 µL of concentrated cells were aliquoted into 5 mL tubes containing 300 mg pre-sterilized 400–625 µm glass beads along with 5 µg of pSS116-VP28, and vortexed at maximum speed for 15 s. High Salt Minimal (HSM) medium supplemented with 0.5% agar was melted and cooled to 42 °C prior to adding 3 mL to each tube followed by immediate pouring onto HSM agar plates. The transformation plates were incubated overnight in the dark and then cultured under 50 µmol photon m^−2^ s^−1^ white light at 25 °C. Successful transformation colonies where photosynthesis had been restored were obtained after 4–6 weeks.

Single colonies were obtained by streaking on fresh HSM plates and cultured for 3–4 weeks before repeating the single-colony isolation. Homoplasmy was monitored by PCR following genomic DNA extraction using a modified CTAB method [24]. To confirm chloroplast DNA extraction, primers rbcL_F and rbcL_R, designed to amplify a 264 bp region of the chloroplast gene *rbcL*, were used as an internal control for PCR analysis [18] (Appendix A). Integration of the *rrnS* transcription cassette carrying the *VP28* fragment was detected using primers VP28_ESP_F and VP28_ESP_R, and primers TN72_F and TN72_R were used to confirm homoplasmy of the transformed chloroplast genome (Appendix A). Once lines were confirmed to be homoplasmic, one line was selected as TN72:VP28, and retained for dsRNA quantification and viral protection studies. A second TN72 transformant line was generated using the pSS116 plasmid using the same method in order to generate a photosynthetically competent control strain. This was named TN72:control and was used as a negative control, together with a previously generated negative control strain named TN72:SR [18].

### 2.5. Double-Stranded RNA Detection and Quantification

TN72:VP28 was inoculated twice into 400 mL TAP medium and cultured by shaking at 100 rpm under continuous light (50 µmol photon m^−2^ s^−1^) at 25 °C. The cultures were harvested by centrifugation at 3000× *g* for 5 min, either at late log phase (day 4) or the 1st day of stationary phase (day 5). Pellets were re-suspended using Trizol solution (Thermo Fisher, Waltham, MA, USA) for RNA extraction to obtain total RNA. Contaminating DNA and single-stranded RNA were removed by DNase I and RNase A treatment, respectively, as per the manufacturer’s instructions to obtain a dsRNA fraction. VP28-specific dsRNA was detected by RT-PCR (RBC Bioscience, Taiwan) using VP28_ESP_F and VP28_ESP_R primers (Appendix A). A pre-digestion of the dsRNA template using RNase III, and a subsequent failure of the RT-PCR reaction was used to demonstrate that the VP28 RNA was indeed double-stranded.

Quantitative RT-PCR (qRT-PCR) was performed using the same primer sequences as for RT-PCR (KAPA Biosystem, Wilmington, MA, USA). Isolated dsRNA was incubated at 95 °C for 5 min and immediately chilled on ice for 2 min prior to addition into the qRT-PCR reaction to ensure denaturation of dsRNA. A series of ten-fold dilutions of purified dsRNA from the bacterial expression was used to generate a standard curve. The amount of dsRNA was calculated according to a formula obtained from the standard curve (R^2^ = 0.958).

### 2.6. Hanging-Bag Culture and Biomass Preparation

Pre-inocula of TN72:VP28 and TN72:SR [18] were prepared by inoculating a full loop of the strain grown on TAP-agar plates into 100 mL TAP medium and shaking under continuous light at 25 °C for 3–4 days. Absorbance at 750 nm was measured and calculated for a volume input into the polythene tubular bags containing 17 L TAP medium to make a starting absorbance at 0.1. The hanging bag photobioreactor was set up according to [25]. Filter-sterilized air was provided from the bottom of the bag and continuous illumination was provided at 100 µE m^−2^ s^−1^ by Osram Lumilux Cool daylight fluorescence tubes. The bag cultures were grown for 5 days at 25 °C with growth monitored by measuring absorbance at 750 nm every 12 h. At harvesting time, chemical flocculation by FeCl_3_ at a final concentration of 2 mM was performed with continuous air supplied for 10–15 min before leaving the bag to stand until the algae had completely precipitated. Microalgal biomass was recovered by centrifugation of the concentrated culture at 3000× *g* for 20 min and the paste was immediately frozen at –20 °C. This was then lyophilized in a freeze-dryer over a period of 24 h, and the dried biomass was kept at –20 °C until further analysis. Double-stranded RNA purification and detection by RT-PCR were performed, as above.

### 2.7. Preparation of Animals and Viral Inoculum

Specific Pathogen Free (SPF) Post larval (PL) 23-day old *Penaeus vannamei* shrimp were purchased from a major hatchery farm (supplier wishes to remain anonymous) and acclimatized in 10 parts per trillion (ppt) salinity artificial seawater for 7 days prior to use as an animal model. The average animal mass and length were approximately 0.03 g and 1.5 cm, respectively, on the first day of experiment. To prepare the WSSV inoculum for the challenge assay, a mixed-gender group of 20 shrimp (10–15 g each) was fed with WSSV-infected homogenized shrimp tissue at a level of 10% of total body mass. Visibly infected shrimps were euthanized and muscle and pleopod tissue collected. WSSV infection was confirmed using the IQ2000 PCR-based detection kit (GeneReach Biotechnology Corp., Taichung, Taiwan) as described in [26] using the WSSV447 primers given in Appendix A. Infected shrimp muscle tissue was homogenized and kept at –20 °C for later use in the oral challenge assay.

### 2.8. Assay for Viral Protection in Shrimp

The experiment was divided into four feeding groups of 100, each representing four replicates of 25 animals. Each replicate was cultured in its own glass tank containing 2 L of 10 ppt artificial seawater with continuous aeration. Groups one and two represented negative and positive controls, respectively, and were fed with commercial feed twice daily at 5% body weight throughout the experiment. Groups three and four were fed with commercial feed supplemented at a 1:1 ratio with dried TN72:SR (negative control [18]) and dried TN72:VP28, respectively, in the same amount as groups one and two. The WSSV challenge was conducted on day four of the experiment. Groups two, three, and four were exposed to oral administration of WSSV-infected tissue at a rate of 50% total body mass (1.25 × 10^8^ WSSV copies/tank). Group one was not exposed. Animal survival was observed daily until group two reached 100% mortality.

### 2.9. Statement of Informed Animal Rights

Experiments related to shrimp used in this study were approved by BT-IACUC, protocol no. BT Animal 8/2562. All procedures performed in studies involving aquatic animals were in accordance with the guidelines of the state government of New South Wales, Australia, for the humane harvesting of fish and crustaceans.

## 3. Results

### 3.1. A New Vector for dsRNA Production in the C. reinhardtii Chloroplast through Convergent Transcription

The p2xTRBL expression vector was designed using two convergent promoters, with each located on opposite strands such that they face toward each other (Figure 1A). For both, we chose to use the promoter from the endogenous *rrnS* gene since this is the most transcriptionally active promoter in the *C. reinhardtii* chloroplast [27]. To terminate transcription, we placed the 3′ untranslated region (3′UTR) from an endogenous gene downstream of each promoter: namely, the *rbcL* 3′UTR for one promoter copy and the *atpB* 3′UTR for the other, as both have been shown to serve as effective terminators of transgene expression in the chloroplast [28]. Between these promoter-terminator pairs, we inserted a gene cassette encoding the monomeric red fluorescent protein (mRFP) flanked by outward-facing restriction sites for the type IIS enzyme *Esp*3I. Expression of this mRFP cassette in *E. coli* gives rise to pink colonies (Figure 1B) and serves as a simple visual reporter for the successful cloning of target DNA into p2xTRBL, as discussed below.

### 3.2. Golden Gate Assembly Allows Quick and Efficient Construction of Chloroplast Transformation Plasmids

In order to target WSSV for dsRNA-mediated gene silencing, a section of the VP28 gene was generated as a 0.3 kb PCR product with an inward-facing *Esp*3I site incorporated at either end such that the 4-base overhangs created following *Esp*3I digestion corresponded to those generated by excision of the mRFP cassette from p2xTRBL. A ‘one-pot’ digestion and ligation of the p2xTRBL and PCR product resulted in efficient directional cloning, as per Golden Gate assembly [29], with the resulting p2xTRBL-VP28 plasmid giving rise to white colonies (Figure 1B). Of 26 white colonies selected, all were found to contain correctly assembled p2xTRBL-VP28 plasmid. As this plasmid has the VP28 DNA under the control of the chloroplast transcriptional elements, both strands should be actively transcribed in the algal chloroplast. However, in order to insert the transcriptional cassette into a neutral site within the plastome, the cassette needs to be flanked with chloroplast DNA elements that allow two homologous recombination events between plasmid and plastome [30]. A second round of Golden Gate cloning was therefore carried out, this time using the type IIS *Bsa*I sites that flank the cassette (Figure 1A) in order to clone it into a new chloroplast integration vector termed pSS116. As before, this plasmid was designed to contain the mRFP cassette as a colorimetric marker, and its replacement with the VP28 cassette to generate pSS116-VP28 was detected as white colonies. Furthermore, the use of different antibiotic selection markers on pSS116 and p2xTRBL (tetracycline and ampicillin resistance, respectively) ensured selection against any white colonies arising from any remaining uncut p2xTRBL-VP28. Once again, all white colonies analyzed contained the correctly assembled plasmid.

### 3.3. The Dual-rrnS Promoter System Is Capable of dsRNA Production in E. coli

The *C. reinhardtii rrnS* promoter has previously been shown to be functional in *E. coli* [31]. The pSS116-VP28 plasmid was therefore transformed into the mutant *E. coli* strain HT115(DE3) that is defective in RNase III [22]. This enzyme specifically degrades double-stranded RNAs and therefore expression in HT115(DE3) allows stable accumulation of transgenic dsRNA. Mid-log phase cultures containing either pSS116-VP28, or pSS116 as a negative control, were used for RNA extraction and isolation of the dsRNA fraction. Analysis of the dsRNA by agarose gel electrophoresis showed a band from the pSS116-VP28 strain migrating to a position close to that of the 0.4 kb DNA marker, whereas no such band was observed for the pSS116 strain (Figure 2A). The predicted size of the *VP28* dsRNA ranges from 365 to ~600 bp depending on where *E. coli* transcription terminates within the chloroplast 3′UTR regions, and whether single-stranded regions at the ends are prone to cleavage (Figure 2B). As dsRNA molecules are expected to migrate similarly to dsDNA under non-denaturing gel conditions, we concluded that the band at 0.4 kb is *VP28* dsRNA. This was confirmed by subsequent RT-PCR analysis of the dsRNA fractions using primers specific to the *VP28* sequence. As shown in Appendix A, an amplicon of the expected size (338 bp) representing just the *VP28* region of the dsRNA was obtained for the fraction from the pSS116-VP28 strain, but not for the pSS116 control.

### 3.4. Generation of C. reinhardtii Transplastomic Lines

The pSS116 chloroplast integration vector is designed with chloroplast flanking sequences such that cloned DNA is targeted into the plastome at a neutral intergenic site between *psbH* and *trnE2* (Figure 1A). Furthermore, a wild-type copy of *psbH* on the right flanking element serves as a native selectable marker, restoring the Δ*psbH* mutant TN72 to phototrophic growth [19]. Chloroplast transformants of TN72 were generated using pSS116-VP28, with seven independent colonies selected for analysis. A pSS116 transformant line was also generated as a negative control. Transformants were restreaked to single colonies twice under selective conditions to drive the polyploid plastome to a homoplasmic state where all copies of the plastome contain the transgenic DNA [30]. Homoplasmy was confirmed by PCR analysis for all eight lines as shown in Figure 3, and one representative transformant line (T_1_) containing the *VP28* cassette was selected for dsRNA investigation together with the negative control transformant (T_C_). These lines were named TN72:VP28 and TN72:control, respectively.

### 3.5. Convergent rrnS Promoters Enhance Production of dsRNA Relative to the Previous Study

In order to investigate whether *VP28* dsRNA accumulated in the TN72:VP28 strain, total RNA was extracted from both TN72:VP28 and TN72:control, and treated with DNase I and RNase A to digest DNA and ssRNA, respectively [18]. The presence of VP28 RNA in the TN72:VP28 preparation was confirmed by RT-PCR with an amplicon of the expected size of 307 bp, whereas no band was seen for the TN72:control (Figure 4A). Furthermore, no amplicon was obtained when the TN72:VP28 RNA template was pre-treated with the dsRNA-specific enzyme, RNase III prior to RT-PCR (Figure 4B), confirming that double-stranded *VP28* RNA was indeed produced in the TN72:VP28 line.

To estimate the amount of VP28 dsRNA and investigate whether the growth phase influenced the level of dsRNA, semi-quantitative RT-PCR was performed on samples harvested after four and five days of culture. These data showed a significantly higher yield of dsRNA from the 5-day culture suggesting accumulation of dsRNA in early stationary phase (Figure 4C). Fully quantitative qRT-PCR was then used on the sample harvested on the 5th day to give a conclusive dsRNA yield. Melt curve analysis using purified dsRNA from the HT115(DE3) strain harboring pSS116-VP28 gave a melting temperature (T_m_) of 82.40 ± 0.32 °C while dsRNA obtained from TN72:VP28 gave a T_m_ of 82.48 ± 0.16 °C. Quantitative analysis gave 41.13 pg *VP28* dsRNA from a 500 ng RNA input, which corresponds to 119 ng of VP28 dsRNA from 1 L of TN72:VP28 culture (OD_750_ at 3.9). In our previous study [18] in which dsRNA from a YHV gene was produced through convergent transcription using promoters from the endogenous *psaA-1* gene, the estimated yield of dsRNA was 16 ng/1 L of culture (OD_750_ at 1.8). While the two yields are not directly comparable given the different dsRNA sequences and OD values, it appears that the new *rrnS* convergent promoter system gives an order of magnitude improvement in dsRNA yield.

### 3.6. Shrimp Fed with the TN72-dsVP28 Line Show ~69% Survival against WSSV

To evaluate the efficiency of dried TN72:VP28 as a feed additive to protect against WSSV infection, a feeding experiment followed by a WSSV challenge assay was performed. Prior to adding it to the feed, the dried algae from TN72:VP28 (and from a TN72 transformant, TN72:SR, lacking any transgene at the *psbH-trnE2* locus [18]) was analyzed by RT-PCR to confirm that the drying process and subsequent storage did not result in the loss of the *VP28* dsRNA. As shown in Appendix A, the dsRNA was readily detected in TN72:VP28, with no RT-PCR product seen for the TN72:SR control.

In the feeding trial, groups of post-larval shrimp were fed for four days with either standard feed or feed mixed 1:1 with dried algae. On day four, the shrimp groups (except for the ‘no virus’ control) were challenged by feeding with WSSV-infected shrimp tissue. As shown in Figure 5, 50% mortality was reached in 4–5 days post-infection (dpi) in the challenged groups that received either no algae or the TN72:SR control supplement, whereas the positive group (no algae, no viral challenge) showed ~90% survival. In contrast, the challenged group that received the feed supplemented with TN72:VP28 showed a marked reduction in mortality compared to the TN72:SR control with ~69% survival at 6 dpi, which then remained at this level. On the last day of the experiment (8 dpi), the survival of both the positive and TN72:SR-supplemented groups was less than 10%, compared to the ~69% survival in the shrimp groups receiving the TN72:VP28 supplement. Statistical analysis of the survival figures is significant at 99% confidence.

## 4. Discussion

The novel vector p2xTRBL together with the transformation vector pSS116 greatly simplifies the construction of dsRNA expressing lines of *C. reinhardtii*, and the use of convergent *rrnS* promoters is a significant improvement over previous work where *psaA* promoters were employed [18]. This is to be expected, as the *rrnS* promoter has been shown to be the most transcriptionally active promoter in the *C. reinhardtii* chloroplast [27,31]. However, it should also be noted that the p2xTRBL cassette represents the first purpose-built *C. reinhardtii* dsRNA expression system. Charoonnart et al. [18] constructed a dsRNA-expressing cassette by modifying the pSRSapI vector to contain convergent *psaA* promoters, but as this vector was designed for protein synthesis, it not only contained unnecessary 5′UTR regions, but only the sense promoter had a corresponding transcriptional terminator. The antisense promoter, on the other hand, would probably drive transcription until a suitable native terminator was reached, adding an extended ssRNA ‘tail’ to the dsRNA as well as wasting cellular resources.

Since chloroplast promoters are prokaryotic in origin, they are often functional in *E. coli*, with the *rrnS* promoter being no exception. This presented a useful opportunity to validate the dsRNA-expressing cassette prior to algal transformation, and also to produce a stock of *VP28* dsRNA to be used as a positive control. Several previous reports have employed convergent bacteriophage T7 promoters to produce dsRNA in *E. coli* [32,33], but stable accumulation of the dsRNA was only possible in the RNase III-deficient *E. coli* strain, HT115(DE3). The amount of dsRNA produced in HT115(DE3) was significant, but was lower than that produced from DNA cassettes expressing a single RNA with sense and antisense sequences that can fold into a hairpin structure [34]. In the case of chloroplast expression, such hairpin dsRNAs have been successfully produced in both *C. reinhardtii* [35] and potato [36]. However, the latter study found that a convergent promoter strategy was superior since the hairpin approach resulted in additional short transcripts that possibly arose because the RNA polymerase was stalling within the extended inverted repeat DNA sequence present in the hairpin cassette [36]. We therefore chose a convergent promoter approach for dsRNA production in the *C. reinhardtii* chloroplast. Furthermore, a marker-free strategy based on the restoration of an endogenous gene was chosen to introduce the dsRNA cassette into the plastome [16]. This avoided creating transgenic strains carrying multiple copies of an antibiotic resistance marker in the polyploid chloroplast. Such strains would be undesirable, particularly when used as part of an aquaculture feed because of the risk of spreading the antibiotic resistance gene to environmental microorganisms via horizontal gene transfer. With our strategy, the only foreign DNA present in the algae is the 307 bp region of the viral gene that is the target for silencing, and this DNA is already present in any environment where viral infection has arisen.

In testing our system, we showed that the TN72:VP28 strain specifically accumulated an RNA fraction that could be amplified by RT-PCR following RNase A treatment (i.e., digestion of single-stranded RNA) but not RNase III treatment (digestion of double-stranded RNA). These results give strong evidence for the production of *VP28* dsRNA in the algal chloroplast. RT-qPCR was used to quantify the *VP28* dsRNA, giving a yield of ~119 ng of VP28 dsRNA from 1 L culture at an OD_750_ of 3.9. Even though this represents a ~10-fold increase over that reported with our previous system, the amount is still much lower than that from a bacterial system. However, the value of the *C. reinhardtii* system is that this microalga is considered safe as a food and feed ingredient [37,38] and can therefore be used as a whole-cell feed. It is also of note that the level of dsRNA was observed to increase from day four to day five of the algal cultivation period. Further work will be needed to confirm this trend, and give a more complete picture of dsRNA accumulation over a full growth period. The length of the dsRNA may also be important: a study of dsRNA production in the chloroplast of potato found that accumulation was negatively correlated with dsRNA length, and suggested that a reduction to ~200 bp might be optimal [39]. It would also be of interest to investigate whether abiotic factors can influence dsRNA accumulation, for example, the effect of light levels, temperature, or media composition.

The drying of the biomass represents an effective method of killing the microalgae, thereby reducing the chance of environmental contamination with a GM species when deployed in aquaculture. Furthermore, the drying produces a powder suitable for formulation into shrimp feed where the dsRNA is potentially stable at room temperature for extended periods. Further analysis is required to establish the exact half-life of the dsRNA under these conditions, together with the dose level needed for effective protection so that a full techno-economic assessment can be made, and a cost-per-dose calculation made. The hanging-bag photobioreactor system offers a low-cost, low-tech strategy for production of the microalgae [25], but it is important to appreciate that harvesting and drying represent key cost determinants in the downstream process stage [40,41]. Increasing the level of dsRNA produced in the chloroplast is therefore an important goal toward minimizing the amount of dried algae per dose.

The *Penaeus vannamei* shrimp is known to be highly susceptible to WSSV throughout its life cycle, but the post-larval (PL) stage of development is considered to be the stage when it is most vulnerable to viral attack according to data collection from hatchery farms and nursery farms throughout Thailand by the Department of Fisheries [42]. It is therefore essential to ensure PL stocks are healthy and disease-free before transferring them to grow-out ponds. While there are challenges with any feed-based vaccine, such as ensuring sufficient stability and availability once added to the pond water and ensuring sufficient dose delivery to all individuals within a large PL shrimp population, the feeding of microalgae expressing dsRNA can clearly reduce or delay the massive mortality following viral infection. If combined with good farm management, this may be able to rescue shrimp from mortality and achieve some profits instead of completely discarding the whole crop.

## Figures and Tables

**Figure 1 microorganisms-11-01893-f001:**
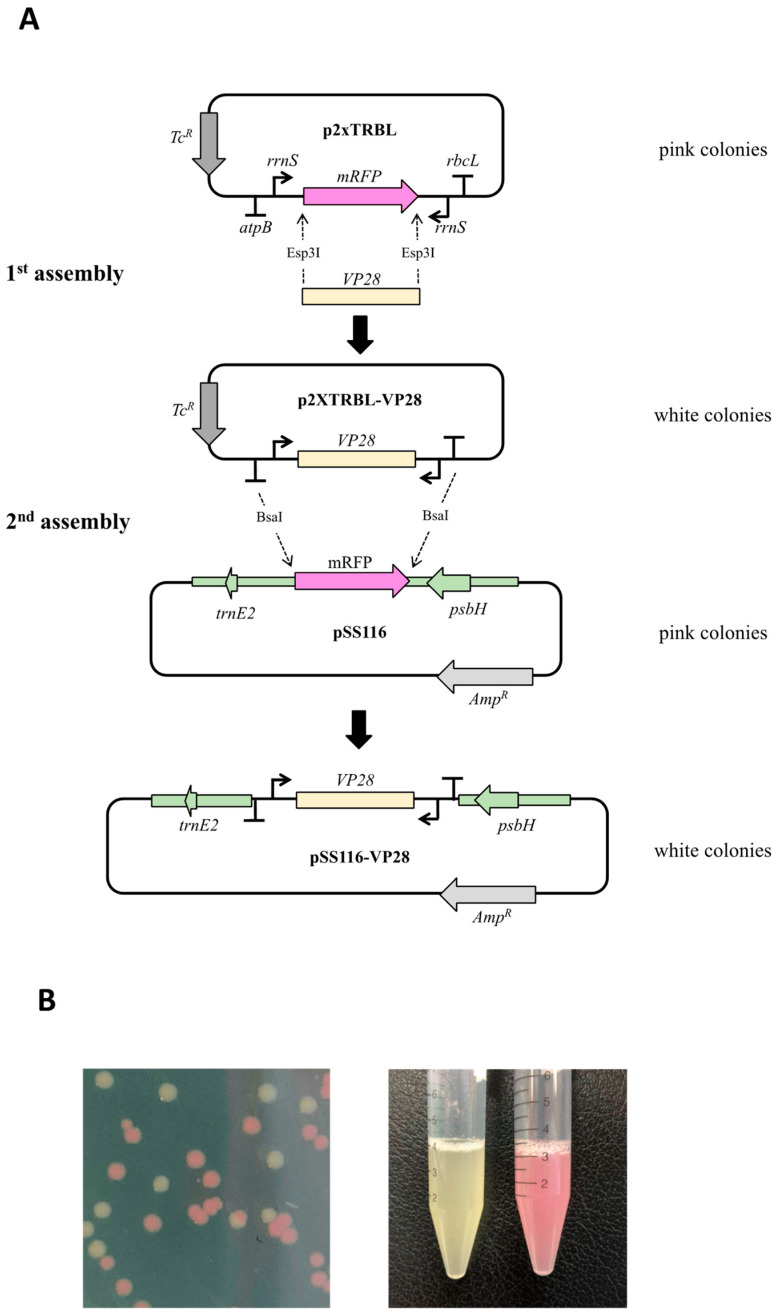
The p2xTRBL vector allows cloning of any DNA fragment between convergent *rrnS* promoters to create a dual transcription cassette: (**A**). Construction of pSS116-VP28 involved a two-step cloning strategy in which the *VP28* DNA was first cloned into p2xTRBL using the type IIS enzyme *Esp*3I and then the dual transcription cassette was cloned into the transformation vector pSS116 using a second type IIS enzyme, *Bsa*I such that the cassette is flanked with chloroplast DNA elements. This allows integration of the cassette into the *psbH-trnE2* intergenic region of the plastome. (**B**). Cloning was aided at each stage by a simple ‘pink/white’ screen where transformant colonies and cultures carrying the correctly assembled plasmid gave a white phenotype.

**Figure 2 microorganisms-11-01893-f002:**
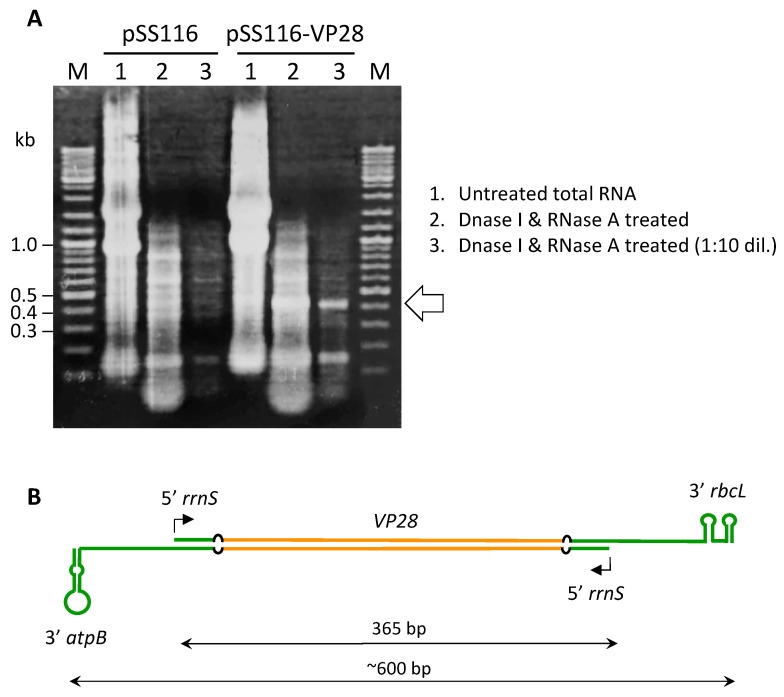
Analysis of RNA extracted from *E. coli* strain HT115(DE3) carrying either pSS116 (negative control) or pSS116-VP28: (**A**). Lanes 1 are total RNA, lanes 2 are the RNA following digestion with both DNase I and RNase A, and lanes 3 are a ten-fold dilution of the digested RNA. The white arrow highlights a DNase I- and RNase A-resistant band that is specific to the pSSP116-VP28 strain and is assumed to be *VP28* dsRNA. (**B**). Approximate sizes and secondary structure of the *VP28* dsRNA assuming that transcription termination occurs in *E. coli* at similar sites within the *atpB* and *rbcL* 3′UTRs as compared to the *C. reinhardtii* chloroplast.

**Figure 3 microorganisms-11-01893-f003:**
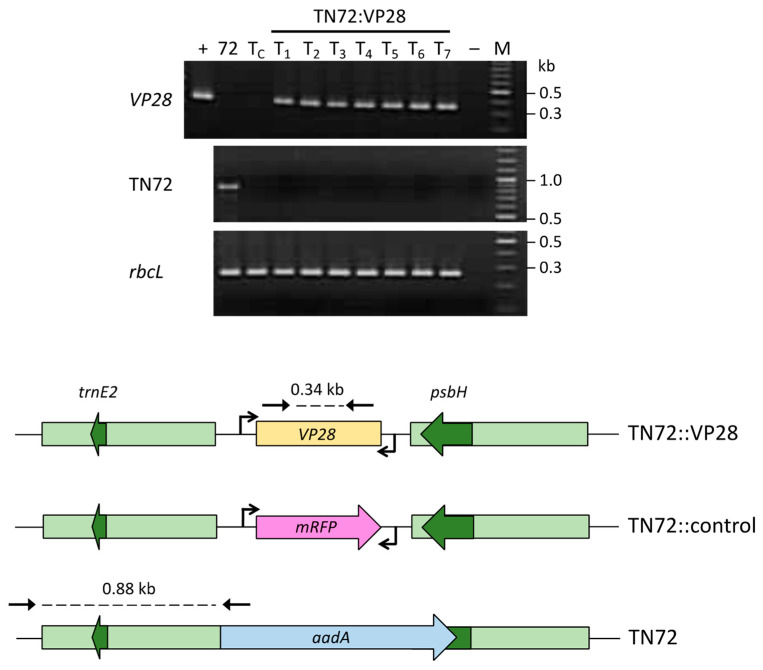
PCR analysis of chloroplast transformants lines. Genomic DNA from seven TN72:VP28 lines (T_1–7_), the control transformant (T_C_), and the untransformed recipient strain TN72 were analyzed using three sets of primers. The VP28 primers (top panel) gave a 0.34 kb product for all seven TN72:VP28 lines, which corresponded to the product generated from the plasmid DNA (+), with no product detected for either TN72 (72), T_C,_ or a ddH_2_O negative control (−). Primers specific for the TN72 plastome (middle panel) gave a 0.88 kb product only for the untransformed strain indicating that all transformed strains were homoplasmic. A third PCR using primers specific to the endogenous *rbcL* gene (lower panel) served as an additional control, with all genomic DNA samples giving a product of 264 bp.

**Figure 4 microorganisms-11-01893-f004:**
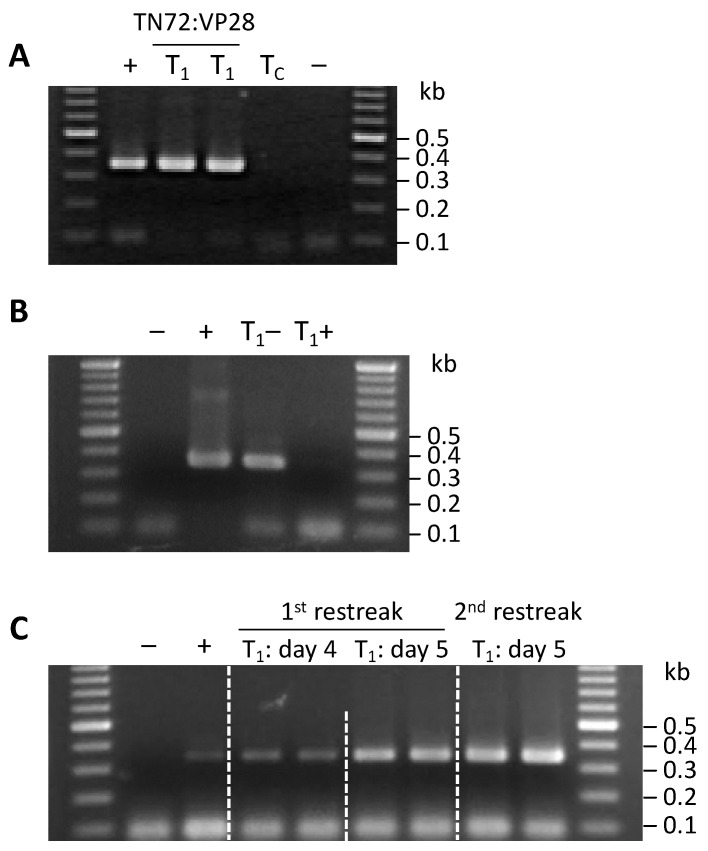
Detection of dsRNA in *C. reinhardtii* strain TN72:VP28: (**A**). RT-PCR analysis of dsRNA isolated from transformant TN72:VP28 (T_1_) yielded a *VP28* amplicon, unlike the TN72:control sample (T_C_). (**B**). Pre-treatment of the dsRNA from TN72:VP28 (T_1_+) with RNase III results in no *VP28* amplicon, in contrast to the untreated dsRNA (T_1_−). (**C**) Semi-quantitative determination of dsRNA yield at different harvesting times showing a marked increase at day 5. For all three panels, the positive (+) and negative (−) controls for the RT-PCR used the dsRNA template from the *E. coli* pSS116-VP28 transformant and ddH_2_O, respectively.

**Figure 5 microorganisms-11-01893-f005:**
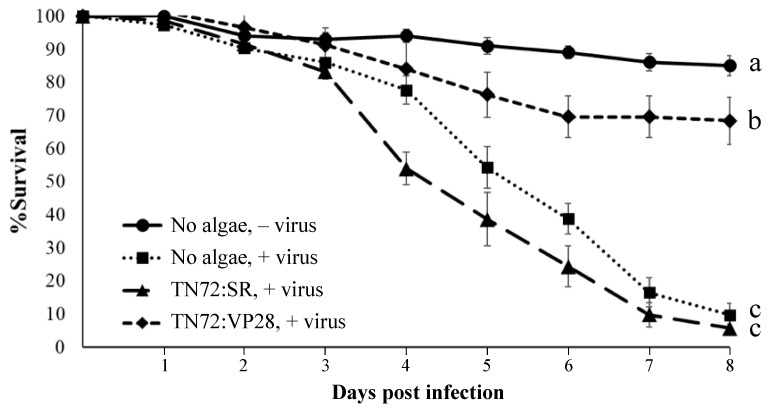
Percentage survival of shrimp during the challenge trial. Groups are: ‘feed only/no viral challenge’ (positive group); ‘feed only/viral challenge’ (negative group); ‘feed + TN72:SR algae/viral challenge’ (−VP28 dsRNA control); ‘feed + TN72:VP28 algae/viral challenge’ (VP28 dsRNA experiment). Bars represent standard error. Letters a, b, and c represent homogeneous subsets grouped by Duncan’s new multiple range test using % survival at 8 dpi.

## Data Availability

Not applicable.

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
