# Peer review of "Transgenic Microalgae Expressing Double-Stranded RNA as Potential Feed Supplements for Controlling White Spot Syndrome in Shrimp Aquaculture"

_microorganisms, 2023, doi:10.3390/microorganisms11081893_

Round 1

Reviewer 1 Report

The manuscript by Charoonnart and co-workers describes the use of RNA interference (RNAi) to disrupt viral gene expression, which may help prevent or treat viral infections in farmed aquatic animals, such as fish and shellfish. In their previous work, the authors had shown that double-stranded RNA (dsRNA) could be produced in the chloroplasts of an edible microalga called Chlamydomonas reinhardtii. This dsRNA was used successfully to manage disease in shrimp. In the current work, the authors have made significant progress in improving the production of this antiviral dsRNA. The subject of their work is critically important, as viral infections in farmed aquatic animals present substantial challenges within the aquaculture industry.

The manuscript is well-written, and the results of the study are robustly supported. The manuscript is suitable for publication following minor revisions.

Minor comments:

1.       In the Materials and Methods section, please specify the manufacturers for the materials used, including bacterial strains. Also, include the concentrations of all antibiotic stock solutions.

2.       Throughout the text, starting with line 102, please decide whether or not to capitalize the "L" in the word "Level." Consistency in formatting will improve the readability of the manuscript.

3.       On line 124, please explain or define the abbreviation "LB+Amp100" to ensure clarity for all readers.

4.       Likewise, please provide the full name for all abbreviations used.

The quality of English is acceptable.

Author Response

The manuscript is well-written, and the results of the study are robustly supported. The manuscript is suitable for publication following minor revisions.

We thank the reviewer for their positive comments.

Minor comments:

  1. In the Materials and Methods section, please specify the manufacturers for the materials used, including bacterial strains. Also, include the concentrations of all antibiotic stock solutions.

We have added these details to the M&M section.

  1. Throughout the text, starting with line 102, please decide whether or not to capitalize the "L" in the word "Level." Consistency in formatting will improve the readability of the manuscript.

We have corrected this so that the "Levels" of Golden Gate assembly are capitalized.

  1. On line 124, please explain or define the abbreviation "LB+Amp100" to ensure clarity for all readers.

This has been defined in the M&M section.

  1. Likewise, please provide the full name for all abbreviations used.

We have done this throughout the manuscript.

Reviewer 2 Report

The manuscript submitted by Charoonnart et al. was designed very well and has an interesting, hot, and original topic. I believe this version of the manuscript is suitable for publication with a few minor corrections listed below. However, it would be much better if the authors have done some histological and immunological assays to evaluate the efficacy and immunogenicity of their approach.   

1-Please, discuss the limitation and difficulties of your approach.

2-How much dried mass did you use for feeding the shrimps? Please provide some data for this.

3-Write C. reinhardtii and E. coli in italics on lines 233 and 240 respectively.

English style and quality are perfect. 

Author Response

The manuscript submitted by Charoonnart et al. was designed very well and has an interesting, hot, and original topic. I believe this version of the manuscript is suitable for publication with a few minor corrections listed below. However, it would be much better if the authors have done some histological and immunological assays to evaluate the efficacy and immunogenicity of their approach.

We thank the reviewer for their positive comments. The histological and immunological assays are technically challenging and are part of our on-going work.

1-Please, discuss the limitation and difficulties of your approach.

We have added some text to the Discussion to highlight potential limitations and difficulties.

2-How much dried mass did you use for feeding the shrimps? Please provide some data for this.

This was detailed on lines 216-217

“…fed with commercial feed twice daily at 5% body weight throughout the experiment.”

3-Write C. reinhardtii and E. coli in italics on lines 233 and 240 respectively.

This has been corrected.